# Reduced macular vessel density and inner retinal thickness correlate with the severity of cerebral autosomal dominant arteriopathy with subcortical infarcts and leukoencephalopathy (CADASIL)

**Chao-Wen Lin[1], Zih-Wei Yang[1], Chih-Hao Chen [2]\*, Yu-Wen Cheng[3], Sung-Chun Tang[2], Jiann-Shing Jeng[2]**

1 Department of Ophthalmology, National Taiwan University Hospital, Taipei, Taiwan, 2 Department of Neurology, National Taiwan University Hospital, Taipei, Taiwan, 3 Department of Neurology, National Taiwan University Hospital Hsin-Chu Branch, Hsinchu, Taiwan

\* antonyneuro@gmail.com

## Abstract

### Background

Cerebral autosomal dominant arteriopathy with subcortical infarcts and leukoencephalopathy (CADASIL), caused by mutations in *NOTCH3*, is the most common cause of hereditary cerebral small vessel disease. Whether it will involve systemic vasculopathy such as retinal vessel remains unknown. Optical coherence tomography angiography (OCT-A) is a noninvasive technique for visualising retinal blood flow. We analysed vessel density and retinal thickness in patients with CADASIL and investigated their correlations with disease severity.

### Methods

This prospective study enrolled 35 patients with CADASIL (59 eyes) and 35 healthy controls (54 eyes). OCT-A was used to measure the vessel density of the macular region and the thickness of retinal layers. Patients with CADASIL were divided into stroke (n = 20) and nonstroke (n = 15) subgroups and underwent cognition and gait speed evaluation. Neuroimaging markers of cortical thickness, white matter hyperintensity, lacunae, and cerebral microbleeds were examined through brain magnetic resonance imaging.

### Results

The OCT-A parameters, including vessel density, were comparable between the patients with CADASIL and the controls. In patients with CADASIL, vessel density in the superficial retinal plexus in the macula as was inner retinal thickness was significantly lower in the stroke than the nonstroke subgroup. Macular vessel density and inner retinal thickness were positively correlated with gait speed, while negatively correlated with number of lacunae.

**Data Availability Statement:** All relevant data are within the paper and its Supporting information files.

**Funding:** This work was supported by National Taiwan University Hospital under the grant number 108-N4163. The funders had no role in study design, data collection and analysis, decision to publish, or preparation of the manuscript.

**Competing interests:** The authors have declared that no competing interests exist.

## Conclusions

OCT-A is potentially a useful tool for evaluating disease severity, ischaemic burden, and neurodegeneration in patients with advanced CADASIL.

## Introduction

Cerebral autosomal dominant arteriopathy with subcortical infarcts and leukoencephalopathy (CADASIL) is one of the most common hereditary cause of stroke and cerebral small vessel disease (SVD). The clinical manifestations usually start from the third to fourth decade and include migraine, recurrent stroke, gait disturbance, and dementia [1,2]. It is caused by mutations of *NOTCH3* gene on chromosome 19 that typically affect highly conserved cysteine residues within the epidermal growth factor–like repeat domain [3]. In brain magnetic resonance imaging (MRI), patients with CADASIL exhibit the typical characteristics of cerebral SVD such as lacunae, white matter hyperintensity (WMH), dilated perivascular spaces, and cerebral microbleeds (CMBs) [1,4]. Neuroimaging features have often been used as surrogate markers for monitoring disease progression and treatment response in studies on cerebral SVD and CADASIL [5,6].

Because CADASIL is a hereditary arteriopathy, nonneurological manifestations and vascular abnormalities outside cerebral arterioles are worthy of investigation. Notably, retinal and cerebral arterioles share similar embryonic, anatomic, and physiological features [7,8]. The deposit of granular osmiophilic material (GOM) in the smooth muscle cells of vessel walls is the pathological hallmark of the arteriopathy in CADASIL [9]. As one study reported, such deposits in the retinal arteries can result in vessel wall thickening [10]. Narrowing of the retinal arterioles and sheathing have been observed in case series of patients with CADASIL [11–14]. In a retrospective study of 15 patients with CADASIL, fluorescein angiography (FA) revealed silent retinal abnormalities such as nerve fiber loss and cotton wool spots [15]. Other studies have noted elevated outer diameters of the retinal arteries as well as arterial wall thickening [16,17]. However, the visual symptoms of CADASIL can be subtle; visual acuity may remain normal [14,15]. In one study, electrophysiological examination revealed dysfunction in the retinal ganglion cells of asymptomatic children of patients with CADASIL [18]. Furthermore, the retinal nerve fiber layer (RNFL) is considered an extension of unmyelinated axons of the brain and constitutes an accessible marker in vivo for neurodegenerative diseases such as CADASIL. However, studies applying optical coherence tomography (OCT) to patients with CADASIL have obtained conflicting results with regard to the thinning or thickening of the RNFL [16,19,20]. Hemodynamic changes in the retinal microvasculature may offer a fresh perspective regarding cerebral SVD and CADASIL. FA and indocyanine green angiography (ICGA) are standard retinal imaging techniques commonly used to visualize retinal and choroidal perfusion, but both are invasive and are two-dimensional modalities that cannot show blood flow in different retinal layers.

OCT-angiography (OCT-A) is a noninvasive technique that can be used to visualize the vasculature in all retinal and choroidal layers at high resolution. This dye-free imaging technique captures the dynamic motion of erythrocytes in vivo. OCT-A outperforms FA and ICGA in assessing retinal microvasculature for the diagnosis of ophthalmologic disease such as diabetic retinopathy, artery and vein occlusion, and glaucoma as well as neurodegenerative diseases [21]. To the best of our knowledge, OCT-A has only been used to evaluate blood flow in patients with CADASIL in one study [22]. In that study, the vessel density of the deep retinal

plexus was significantly lower in patients with CADASIL than in healthy controls, possibly reflecting pericyte dysfunction in the retinal capillaries. However, the severity of cerebral SVD on brain MRI was not correlated with any OCT-A parameters. Therefore, in the present study, we investigated whether OCT-A parameters can constitute a window through which to observe the impacts of SVD on patients with CADASIL. We especially focused on the relationship between the parameters of OCT-A and cognitive, gait function, and MRI markers in patients with CADASIL.

## Methods

### Participant enrolment and assessment

This study, which adhered to the tenets of the Declaration of Helsinki and was approved by the Research Ethics Committee of National Taiwan University Hospital (NTUH-REC No. 201807044RIND), was conducted at National Taiwan University Hospital from March 2019 to January 2021. Written informed consent was obtained from all the participants.

Patients with clinical and neuroimaging results suggestive of cerebral SVD were screened for *NOTCH3* mutation. The initial presentations of these patients included stroke, cognitive dysfunction, gait disturbance, or headache. In Taiwan, p.R544C in exon 11 of the *NOTCH3* gene accounts for more than 70% of cases of CADASIL [23]. Therefore, all patients in the present study were initially screened for the *NOTCH3* p.R544C mutation. If no mutation was detected, complete sequencing of the *NOTCH3* exons was performed [24]. The patients with confirmed *NOTCH3* mutations were enrolled.

The clinicodemographic characteristics recorded comprised age, sex, hypertension, diabetes mellitus, dyslipidemia, smoking history, alcohol use, headache, dementia, and history of stroke. Stroke was defined as an acute episode of focal neurological dysfunction lasting longer than 24 h with corresponding neuroimaging evidence of cerebral infarction or hemorrhage. Because patients with CADASIL and history of stroke have higher rates of stroke recurrence and worse outcomes [25], the participants were further divided into stroke and nonstroke subgroups. Global cognitive function was evaluated by using the Mini-Mental State Examination (MMSE). The processing speed index subscale from the Wechsler Adult Intelligence Scale-Third Edition and verbal fluency test were used to measure the executive function and working memory, with higher score indicating better performance. Gait function was assessed using a standardized 4-m walk test, with gait speed recorded in m/s [26]. A faster speed reflected better motility function.

Age-matched healthy controls without diabetes, hypertension, coronary artery disease, stroke, dementia, and other neurological or psychiatric disease were recruited and evaluated by the same ophthalmologist (C.W.L.). All enrolled patients and controls underwent a complete ophthalmic evaluation of best-corrected visual acuity, tonometry, autorefractometry, and fundus ophthalmoscopy as well as the slit-lamp examination. Visual field was tested using automated perimetry with a central 30–2 threshold test (Humphrey 740i, Zeiss Meditec, Germany). The exclusion criteria for patients and controls were myopia of >5 diopters or hyperopia of >3 diopters, glaucoma, optic neuropathy, retinopathy (including diabetic retinopathy, hypertensive retinopathy, age-related macular degeneration, and epiretinal membrane), retinal vascular diseases, dense cataract, neurodegenerative disorders such as Parkinsonism or Alzheimer's disease, or any other conditions that could potentially lead to abnormal OCT-A results.

### Optical coherence tomography angiography

OCT-A imaging was performed using a spectral domain OCT-system (AngioVue, RTVue XR Avanti, Optovue, Fremont, CA, USA) after pupillary dilation. The average peripapillary RNFL

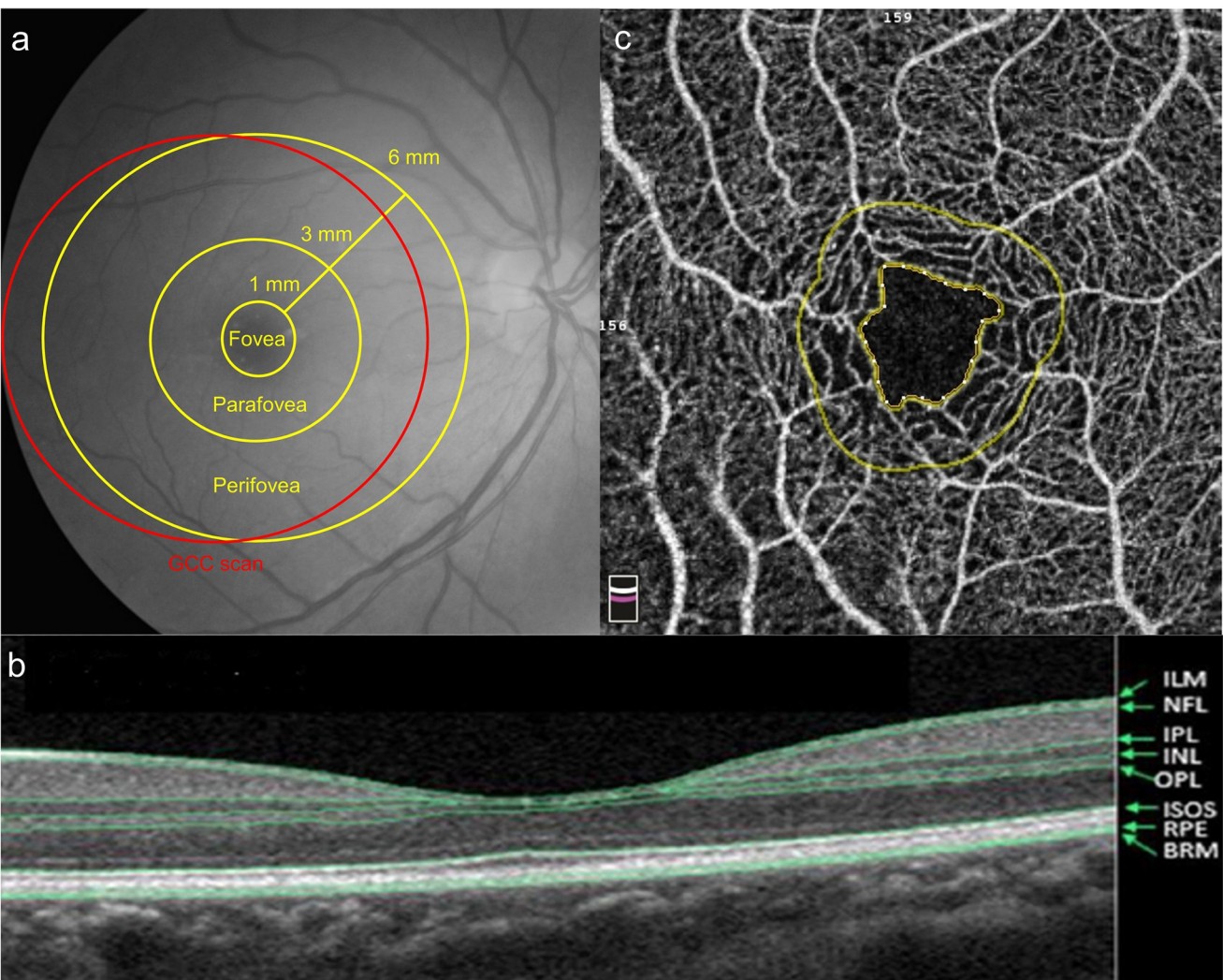

**Fig 1.** (a) The software of optical coherence tomography angiography (OCTA) divided the macula into the fovea, parafovea, and perifovea. The foveal, parafoveal, and perifoveal regions were defined as a circle of 1 mm, 3 mm and 6 mm respectively. Ganglion cell complex (GCC) scan was measured within a circular macular area (6 mm in diameter) that was centered 1 mm temporal to the fovea. (b) The retinal structure showed in OCTA. The inner retina was bounded from the inner limiting membrane (ILM) to the inner plexiform layer (IPL). (NFL: Nerve fiber layer; INL: Inner nuclear layer; OPL: Outer plexiform layer; ISOS: Inner segment / outer segment of photoreceptor; RPE: Retinal pigment epithelium; BRM: Bruch's membrane) (c) The foveal avascular zone (FAZ) was the area inside the inner circle, which was automatically defined by OCTA.

thickness was evaluated along a circle 3.45 mm in diameter that was centered on the optic disc. The average thickness of the ganglion cell complex (GCC), which consists of the RNFL, ganglion cell layer, and inner plexiform layer (IPL), was measured within a circular macular area (6 mm in diameter) that was centered 1 mm temporal to the fovea (Fig 1a). Inner retinal thickness of the macular region was also measured. The inner retina was bounded from the inner limiting membrane to the IPL (Fig 1b). The foveal, parafoveal, and perifoveal regions were defined as a circle of 1 mm, 3 mm and 6 mm respectively according to the standard Early Treatment Diabetic Retinopathy Study grid (Fig 1a). Vessel density was calculated as the percentage area occupied by flowing blood vessels in the segmented region. With a $3 \times 3$ mm$^2$ field of view centered on the fovea, the foveal avascular zone (FAZ) (Fig 1c), vessel density within the superficial vascular complex (VDsM) and deep vascular complex (VDdM) were

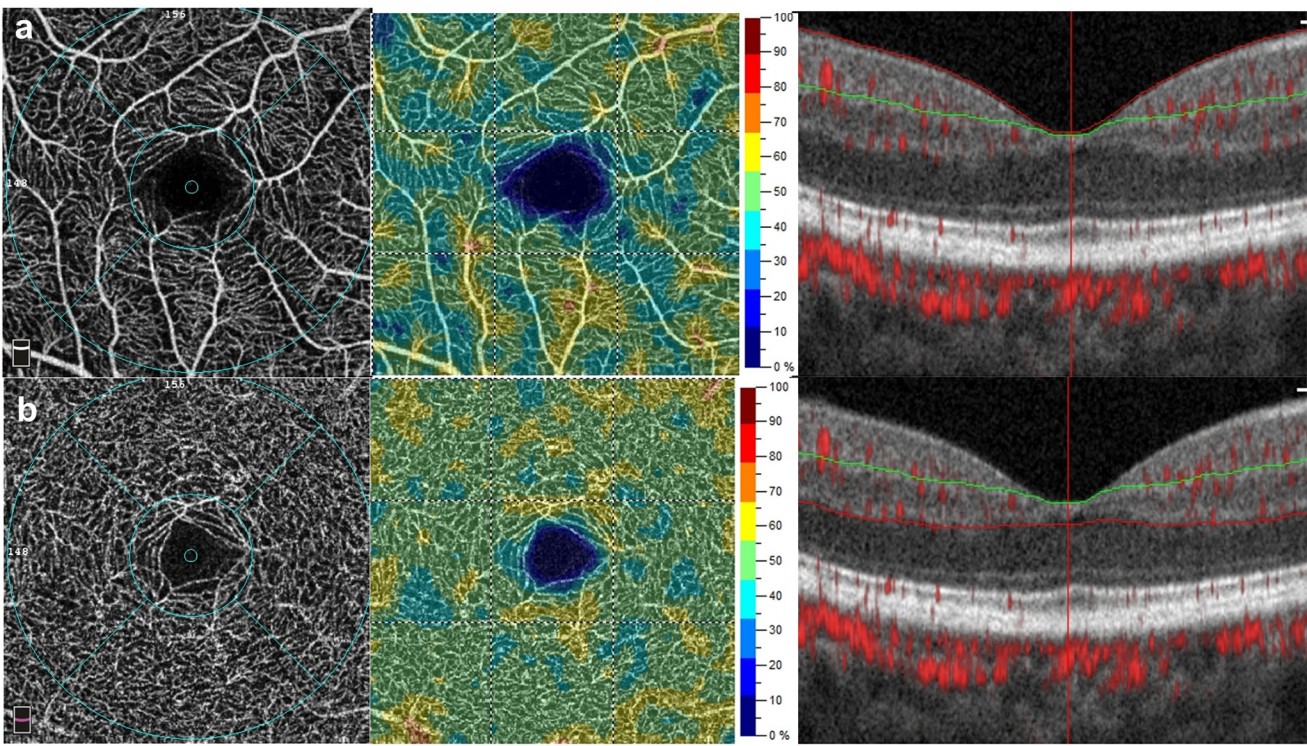

**Fig 2. Representative optical coherence tomography angiography images of the macular region in a patient with CADASIL.** (a) Vessel density measurement in the superficial vascular complex. (b) Vessel density measurement in the deep vascular complex. (Left) The middle and outer circles are the foveal and parafoveal regions, respectively. (Middle) Vessel density map of the superficial or deep vascular complex. (Right) The green lines and red lines on the b-scan constitute the segmentation boundaries for the measurements.

determined using device software. Representative macular scanning images from OCT-A in a patient with CADASIL are shown in Fig 2. Automatic segmentation was verified by two ophthalmologists (C.W.L. & Z.W.Y.). Images of insufficient quality (scan quality < 7/10 or signal strength index < 50) or affected by artifacts were excluded from analysis.

## Neuroimaging assessment

Patients with CADASIL underwent the same 1.5-T brain MRI scanner to evaluate the extent of SVD and any evidence of previous or recent stroke. The scanning sequences included a high-resolution T1-weighted volumetric scan, T2 and fluid-attenuated inversion recovery (FLAIR)-T2 scans for evaluation of WMH and detection of lacunae, susceptibility-weighted imaging for detection of CMBs, and magnetic resonance angiography for intracranial vessels.

WMH were segmented by the lesion growth algorithm as implemented in the Lesion Segmentation Tool (LST) toolbox version 3.0.0 (www.statistical-modelling.de/lst.html) for Statistical Parametric Mapping [27]. A FLAIR sequence was used for lesion segmentation, while a T1 image was used for reference of registration. The final segmented lesions from the output of LST were visually screened for accuracy, and the volumes of WMH were expressed in $cm^3$. To measure the global gray matter atrophy, the mean cortical thickness was quantified on T1-weighted structural MRI scans using the pipeline and output from the FreeSurfer software version 7.2.0 (http://surfer.nmr.mgh.harvard.edu/) [28]. Lacunae were defined as round or ovoid fluid-filled cavities between 3 and 15 mm in diameter in the subcortical area on the FLAIR images. CMBs were defined as presence of homogeneous round areas of signal loss less

than 10 mm in diameter on the susceptibility-weighted imaging. Numbers of lacunae and CMBs were calculated.

## Statistical analysis

All eyes for which image quality was acceptable were subjected to analysis. Therefore, linear mixed effect models were used to test the differences in all eyes, with the individual patient as the repeated unit and age and sex adjusted as fixed effects. First, the parameters of OCT-A in patients with CADASIL and control groups were compared. Next, the parameters of OCT-A in patients with CADASIL in the stroke and nonstroke subgroups were compared, using linear mixed effect models described above. Clinicodemographic characteristics, neuroimaging features, cognitive and gait function in patients with and without stroke were compared using the Wilcoxon rank sum test or chi-square test. Finally, Spearman's rank sum tests were applied to examine whether the parameters of OCT-A results were correlated to the cognition and gait, and neuroimaging markers, respectively. All correlation coefficients were adjusted for age and sex, and the 95% confidence intervals were calculated. The statistical significance level was $P < 0.05$. All analyses were performed using the SAS software, Version 9.4 (SAS Institute Inc., Cary, NC, USA).

## Results

### Participants

Analyses were conducted on 59 eyes of 35 patients with CADASIL and 54 eyes of 35 healthy controls. Overall, *NOTCH3* p.R544C mutation was detected in 31 patients with CADASIL (88.6%), and other patients had *NOTCH3* p.C155Y, p.R141C, or p.R332C mutations. Among the twenty patients (57.1%) in the stroke subgroup, 12, 5, and 3 had experienced ischaemic, hemorrhagic, and both ischaemic and hemorrhagic stroke, respectively. None had had stroke in the posterior cerebral artery territory that may interfere with any ophthalmologic evaluation.

### Parameters of OCT-A results in the CADASIL and control groups, and stroke and nonstroke subgroups in CADASIL

Table 1 presents a summary of the demographic characteristics and parameters of OCT-A in the CADASIL and control groups. After adjustment for age and sex, the OCT-A results were comparable between the patients and controls.

The 35 patients with CADASIL were further divided into the stroke and nonstroke subgroups, which comprised 20 and 15 individuals, respectively. The patients in the stroke subgroup were borderline older, had worse cognition and gait speed, thinner cortical thickness and higher prevalence of CMBs (Table 2). There was no significant difference in the prevalence of comorbidity such as hypertension, diabetes, or dyslipidemia among two subgroups.

Table 3 presents a summary of the age- and sex-adjusted OCT-A results in the stroke and nonstroke subgroups. Among the OCT-A results, VDsM in the fovea (13.43±5.93% vs 19.09 ±6.49%, $P = 0.027$), VDsM (45.66±3.72% vs 49.10±4.48%, $P = 0.024$) in the parafovea, and GCC thickness (94.16±6.09 μm vs 98.96±4.36 μm, $P = 0.011$) were significantly lower in the stroke group, as were inner retinal thickness in the foveal, parafoveal, and perifoveal regions.

### Correlations between parameters of OCT-A and cognition, gait and neuroimaging markers

The age- and sex- adjusted correlations between the OCT-A results and the cognition and gait are shown in Table 4, and the correlation coefficients were also plotted in Fig 3a. Verbal

**Table 1. Comparison of demographic characteristics and parameters of OCT-A in the CADASIL and control groups.**

| | CADASIL | Control | P-value |
|---|---|---|---|
| Case (eye) number | 35 (59) | 35 (54) | |
| Age (years) | 59.6±9.4 | 60.9±12.1 | 0.61 |
| Sex (M:F) | 20:15 | 17:18 | 0.47 |
| Refraction status (diopter) | -1.28±0.40 | -1.08±0.40 | 0.73 |
| **Parameters of OCT-A** | | | |
| FAZ (mm$^2$) | 0.30±0.11 | 0.28±0.11 | 0.44 |
| VDsM, fovea (%) | 16.02±6.76 | 17.59±6.51 | 0.22 |
| VDsM, parafovea (%) | 47.23±4.40 | 47.05±3.97 | 0.96 |
| VDdM, fovea (%) | 31.30±6.89 | 32.23±8.29 | 0.36 |
| VDdM, parafovea (%) | 51.32±4.18 | 51.29±3.66 | 0.84 |
| RNFL (μm) | 101.78±7.85 | 97.50±9.75 | 0.10 |
| GCC (μm) | 96.36±5.85 | 94.72±6.39 | 0.28 |
| IRT, fovea (μm) | 70.14±12.75 | 69.19±12.27 | 0.97 |
| IRT, parafovea (μm) | 125.73±10.32 | 124.50±8.68 | 0.82 |
| IRT, perifovea (μm) | 111.42±6.41 | 108.39±6.33 | 0.11 |

Data are expressed as means ± standard deviations. P value were age- and sex-adjusted.

Abbreviations: OCT-A, optical coherence tomography angiography; FAZ, foveal avascular zone; VDsM, vessel density in the superficial plexus of the central macula; VDdM, vessel density in the deep plexus of the central macula; RNFL, peripapillary retinal nerve fiber layer; GCC, macular ganglion cell complex; IRT, inner retinal thickness.

fluency score was positively correlated with VDsM in the parafovea (Spearman's ρ = 0.62). Gait speed was positively correlated with VDsM in the parafovea (ρ = 0.42), RNFL (ρ = 0.42), and inner retinal thickness in the parafovea (ρ = 0.36).

For the correlations between the parameters of OCT-A and neuroimaging markers (Table 5, Fig 3b), mean cortical thickness was positively correlated with inner retinal thickness in the parafovea and perifovea regions (ρ = 0.37 and 0.46, respectively). As an index of ischaemic burden, the number of lacunae was positively correlated with the FAZ (ρ = 0.39), while negatively correlated with VDdM in the fovea (ρ = -0.34) and inner retinal thicknesses in the fovea (ρ = -0.36). No significant correlations between the OCT-A results and volume of WMH or number of CMBs were observed.

## Discussion

Ocular presentations can be crucial signs of various systemic diseases, such as hypertension, diabetes, leukemia, and neurodegenerative diseases. Retinopathy severity may also be correlated with the staging of systemic diseases. OCT-A is a noninvasive technique for analyzing the perfusion in the macular region. Because cerebral and retinal vessels have some commonalities, OCT-A may constitute a method for predicting CADASIL severity and even prognosis. To the best of our knowledge, this is the second study to use OCT-A to evaluate patients with CADASIL and the first to use OCT to measure retinal thickness in this patient population. The present study has the largest number of patients with CADASIL (n = 35) of all OCT studies on individuals with this condition. We discovered that the macular vessel density in the superficial retinal plexus was significantly lower in the stroke subgroup than in the nonstroke subgroup, as was the inner retinal thickness.

**Table 2. Comparison between the stroke and nonstroke subgroups in CADASIL.**

|  | Stroke | Nonstroke | P-value |
|---|---|---|---|
| Case (eye) number | 20 (32) | 15 (27) |  |
| Age | 62.5±9.5 | 55.8±7.9 | 0.05 |
| Male sex | 12 (60%) | 8 (53%) | 0.69 |
| Refraction status (diopter) | -1.46±0.55 | -1.19±0.62 | 0.75 |
| **Clinical variables** |  |  |  |
| Hypertension | 11 (55.0%) | 6 (40.0%) | 0.38 |
| Diabetes mellitus | 6 (30.0%) | 2 (13.3%) | 0.42 |
| Dyslipidemia | 10 (50.0%) | 6 (40.0%) | 0.73 |
| Smoking habit | 8 (40.0%) | 2 (13.3%) | 0.13 |
| Dementia | 12 (60.0%) | 6 (40.0%) | 0.31 |
| Headache | 3 (15.0%) | 4 (26.7%) | 0.43 |
| **Cognition and gait** |  |  |  |
| MMSE | 28 (26–29) | 29 (28–30) | **0.01** |
| Processing speed index | 86 (81–100) | 104 (99–108) | **0.007** |
| Verbal fluency | 28 (22–34) | 42 (31–51) | **0.01** |
| Gait Speed, m/sec | 0.81±0.30 | 1.08±0.20 | **0.008** |
| **Neuroimaging variables** |  |  |  |
| WMH volume, cm$^3$ | 40.6 (23.5–53.5) | 36.9 (7.9–60.7) | 0.61 |
| Cortical thickness, mm | 2.38±0.12 | 2.44±0.08 | **0.02** |
| Lacunae number | 5 (1–11) | 1 (0–6) | 0.07 |
| CMBs number | 10 (4–39) | 6 (0–12) | 0.27 |
| CMBs presence (%) | 18 (90.0%) | 9 (60.0%) | **0.04** |

Data are expressed as means ± standard deviations, medians (interquartile ranges), or numbers (percentages).

Numbers in bold indicate statistical significance.

Abbreviations: MMSE, Mini-Mental State Examination; WMH, white matter hyperintensity; CMBs, cerebral microbleeds.

**Table 3. Comparison of the parameters of OCT-A between the stroke and nonstroke subgroups in CADASIL, with age- and sex-adjusted.**

| Adjusted means | Stroke | Nonstroke | P-value |
|---|---|---|---|
| Number of cases (eyes) | 20 (32) | 15 (27) |  |
| FAZ (mm$^2$) | 0.33±0.10 | 0.26±0.11 | 0.104 |
| VDsM, fovea (%) | 13.43±5.93 | 19.09±6.49 | **0.027** |
| VDsM, parafovea (%) | 45.66±3.72 | 49.10±4.48 | **0.024** |
| VDdM, fovea (%) | 28.87±5.87 | 34.19±6.99 | 0.050 |
| VDdM, parafovea (%) | 51.50±3.99 | 51.12±4.47 | 0.307 |
| RNFL (μm) | 100.34±8.80 | 103.48±6.27 | 0.423 |
| GCC (μm) | 94.16±6.09 | 98.96±4.36 | **0.011** |
| IRT, fovea (μm) | 66.03±10.16 | 75.00±13.94 | **0.030** |
| IRT, parafovea (μm) | 121.09±10.03 | 131.22±7.74 | **0.004** |
| IRT, perifovea (μm) | 108.97±6.97 | 114.33±4.17 | **0.025** |

Data are expressed as means ± standard deviations. P value were age- and sex-adjusted.

Numbers in bold indicate statistical significance.

Abbreviations: OCT-A, optical coherence tomography angiography; FAZ, foveal avascular zone; VDsM, vessel density in the superficial plexus of the central macula; VDdM, vessel density in the deep plexus of the central macula; RNFL, peripapillary retinal nerve fiber layer; GCC, macular ganglion cell complex; IRT, inner retinal thickness.

**Table 4. Correlations between parameters of OCT-A and cognition and gait.**

|  | MMSE | Processing speed | Verbal fluency | Gait Speed |
|---|---|---|---|---|
| FAZ | -0.28 (0.11) | -0.11 (0.56) | -0.02 (0.91) | -0.23 (0.20) |
| VDsM, fovea | 0.28 (0.11) | 0.27 (0.16) | 0.08 (0.66) | 0.33 (0.07) |
| VDsM, parafovea | 0.15 (0.42) | 0.28 (0.14) | **0.62 (0.0002)** | **0.42 (0.01)** |
| VDdM, fovea | 0.28 (0.12) | 0.24 (0.20) | 0.07 (0.70) | 0.26 (0.15) |
| VDdM, parafovea | 0.07 (0.71) | 0.001 (0.99) | -0.02 (0.93) | -0.01 (0.94) |
| RNFL | -0.17 (0.35) | 0.14 (0.48) | 0.25 (0.18) | **0.42 (0.02)** |
| GCC | -0.02 (0.92) | 0.09 (0.64) | 0.22 (0.24) | 0.28 (0.12) |
| IRT, fovea | 0.03 (0.86) | 0.10 (0.59) | -0.03 (0.87) | 0.25 (0.16) |
| IRT, parafovea | -0.07 (0.69) | 0.13 (0.49) | 0.18 (0.32) | **0.36 (0.04)** |
| IRT, perifovea | -0.18 (0.31) | 0.14 (0.46) | 0.16 (0.40) | 0.18 (0.33) |

Data are expressed as Spearman's ρ (P value), after adjustment of age and sex. Numbers in bold indicate statistical significant.

Abbreviations: OCT-A, optical coherence tomography angiography; MMSE, Mini-Mental State Examination; WMH, white matter hyperintensity; CMBs, cerebral microbleeds; FAZ, foveal avascular zone; VDsM, vessel density in the superficial plexus of the central macula; VDdM, vessel density in the deep plexus of the central macula; RNFL, peripapillary retinal nerve fiber layer; GCC, macular ganglion cell complex; IRT, inner retinal thickness.

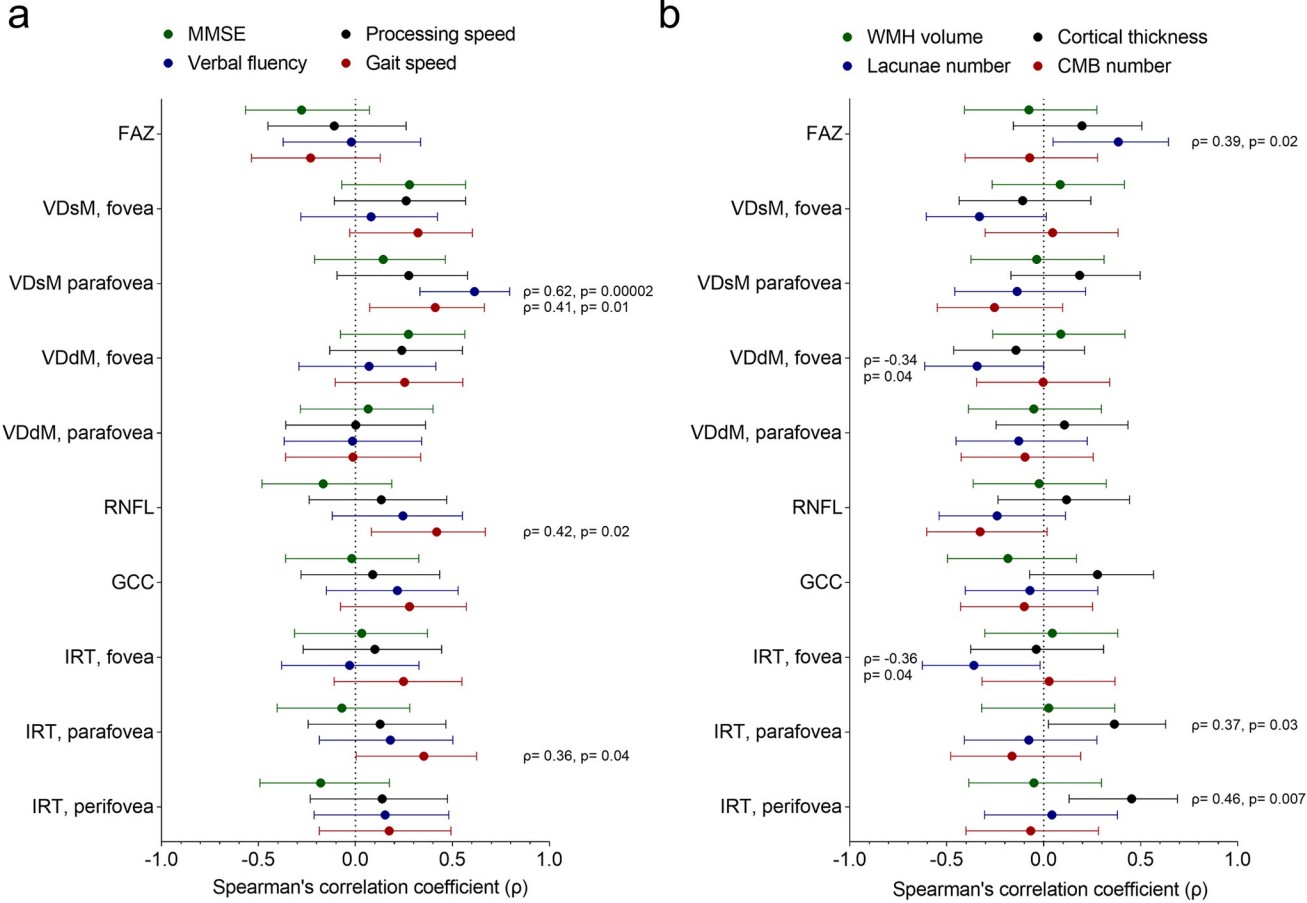

**Fig 3.** (a) The correlation coefficients between the parameters of OCT-A and cognition and gait markers. (b) The correlation coefficients between the parameters of OCT-A and neuroimaging markers. The central dot and the error bar represent the correlation estimates and its 95% confidence intervals.

**Table 5. Correlations between parameters of OCT-A and neuroimaging markers.**

|                  | WMH volume    | Cortical thickness | Lacunae number    | CMBs number     |
|------------------|---------------|--------------------|-------------------|-----------------|
| FAZ              | -0.08 (0.67)  | 0.20 (0.26)        | **0.39 (0.02)**   | -0.07 (0.69)    |
| VDsM, fovea      | 0.09 (0.63)   | -0.11 (0.54)       | -0.34 (0.06)      | 0.05 (0.80)     |
| VDsM, parafovea  | -0.04 (0.84)  | 0.19 (0.29)        | -0.14 (0.44)      | -0.26 (0.15)    |
| VDdM, fovea      | 0.09 (0.62)   | -0.15 (0.42)       | **-0.34 (0.04)**  | -0.003 (0.99)   |
| VDdM, parafovea  | -0.05 (0.77)  | 0.11 (0.55)        | -0.13 (0.47)      | -0.10 (0.59)    |
| RNFL             | -0.02 (0.90)  | 0.12 (0.51)        | -0.24 (0.17)      | -0.33 (0.06)    |
| GCC              | -0.19 (0.30)  | 0.28 (0.11)        | -0.07 (0.69)      | -0.10 (0.57)    |
| IRT, fovea       | 0.05 (0.80)   | -0.04 (0.83)       | **-0.36 (0.04)**  | 0.03 (0.87)     |
| IRT, parafovea   | 0.03 (0.88)   | **0.37 (0.03)**    | -0.08 (0.67)      | -0.17 (0.36)    |
| IRT, perifovea   | -0.05 (0.78)  | **0.46 (0.007)**   | 0.04 (0.81)       | -0.07 (0.71)    |

Data are expressed as Spearman's ρ (P value), after adjustment of age and sex. Numbers in bold indicate statistical significant.

Abbreviations: OCT-A, optical coherence tomography angiography; MMSE, Mini-Mental State Examination; WMH, white matter hyperintensity; CMBs, cerebral microbleeds; FAZ, foveal avascular zone; VDsM, vessel density in the superficial plexus of the central macula; VDdM, vessel density in the deep plexus of the central macula; RNFL, peripapillary retinal nerve fiber layer; GCC, macular ganglion cell complex; IRT, inner retinal thickness.

Deposits of GOM in the basal lamina of the smooth muscle of small vessels are pathogno-monic for CADASIL [9]. A case report on patients with CADASIL observed thickening of arterial walls with fibrosis in a histopathological examination as well as pericyte degeneration [10]. However, the choroid was not found to be affected. Other studies on CADASIL have indicated increased outer diameters of retinal vessels [16,17]. Because retinal vessels mainly run within the nerve fiber layer, the thickening of retinal vessels may result in thickening of the RNFL and the inner retina, particularly in the peripapillary and perifoveal regions, which contain larger retinal vessels. By contrast, another study reported RNFL thinning in patients with ischaemic stroke [29], which may be a consequence of microinfarct in the RNFL. Trans-neuronal retrograde degeneration may also play a role [30]. In sum, the thickness of the inner retina and RNFL in patients with CADASIL may be affected by retinal vessel diameter and ischaemic changes in nerve fibers. These two factors may explain the conflicting results on RNFL thickness obtained in previous studies [16,19,20] and the comparable inner retinal thickness of the patients and controls in the present study. The difference between the stroke and nonstroke subgroups in macular GCC thickness was more significant than that in peripa-pillary RNFL thickness; this is ascribable to the larger retinal vessels in the peripapillary region.

In contrast to the report of significantly lower macular vessel density in patients with CADASIL compared with that in healthy controls by Nelis *et al.* [22], we observed no signifi-cant difference in vessel density between the CADASIL and control groups. Vessel density in the superficial plexus of the foveal and parafoveal regions was significantly lower in the patients with stroke than those without stroke. The difference in vessel density in the deep reti-nal plexus was not significant. A study on retinal structural and microvascular alterations in individuals with acute ischaemic stroke of various subtypes noted reduced vessel density, and the deep microvascular network was more sensitive to ischaemic stroke than was the superfi-cial microvascular network [31]. However, another study displayed that decreased vessel den-sity of the superficial retinal plexus could be a biomarker of SVD [32]. Besides, the superficial retinal plexus contains larger vessels than the deep retinal plexus and thus may be more strongly affected by GOM deposits, resulting in arteriolar attenuation. The present OCT-A results indicate an association between vessel density and disease severity. Vessel density may decrease with disease progression and the development of ischaemic stroke. Therefore,

macular vessel density in OCT-A could be a marker of advanced severity of CADASIL rather than an early manifestation.

In the last part of this study, we examined the correlations between the parameters of OCT-A and the clinical and neuroimaging variables. As mentioned, gait speed was correlated with the vessel density of superficial retinal plexus, RNFL, and inner retinal thickness in the parafovea. Moderate correlation was also found between verbal fluency and the vessel density of superficial retinal plexus. These findings were in accordance to a recent study showing the association between the macular vessel density and the performance of multiple cognitive domains [32]. The correlations between OCT-A and gait speed in our study, which has not been reported before, also suggested that OCT-A may be applicable as a functional assessment in patients with hereditary SVD. The number of lacunae was correlated with the size of FAZ, foveal vascular density, and foveal inner retinal thickness. The foveal values in the OCT-A results may also reflect the evidence of ischaemia in neuroimaging. The positive correlations between the mean cerebral cortical thickness and retinal thickness supported the hypothesis of using OCT as a tool to predict neurodegeneration in the brain [33]. Despite being a hallmark of CADASIL, severity of WMH was not correlated with any parameters of OCT-A. Due to the exploratory nature of our analysis, these results could serve as preliminary findings implicated for further studies.

This study has some limitations. First, the number of patients with CADASIL was relatively small. CADASIL is a rare disease, and patients' willingness to undergo detailed ophthalmology examinations may be compromised in the advanced stage. Second, we measured only refraction status, not axial length. Extremely high axial length may affect imaging analysis. We did exclude individuals with high myopia or hyperopia, however, and refraction status was comparable between the CADASIL group (comprising the stroke and nonstroke subgroups) and the controls. Third, we did not directly measure retinal vessel diameter; the inference that the outer diameter of the retinal vessels of patients with CADASIL is greater was based on that presented in previous studies [16,17]. Further investigation involving projection-resolved OCT-A may provide more detailed information on the retinal microvasculature and aid in elucidation of the pathophysiological mechanism of CADASIL [34]. Finally, this study focused on CADASIL, a special form of hereditary SVD. The generalizability of our findings to patients with sporadic SVD may be limited.

In conclusion, vessel density in the superficial plexus of the foveal and parafoveal regions was significantly lower in the stroke subgroup than in the nonstroke subgroup of CADASIL patients, as was central macular inner retinal thickness. Furthermore, macular vascular density and inner retinal thickness were correlated with gait speed and the number of lacunae. The present findings suggest that OCT-A is potentially a convenient tool for the noninvasive determination of disease severity and neurodegeneration in patients with advanced CADASIL.

## Supporting information

**S1 Dataset. CADASIL OCT dataset.**
(XLSX)

## Author Contributions

**Conceptualization:** Chao-Wen Lin, Chih-Hao Chen.

**Data curation:** Chao-Wen Lin, Zih-Wei Yang, Yu-Wen Cheng.

**Formal analysis:** Chao-Wen Lin, Chih-Hao Chen.

**Funding acquisition:** Chao-Wen Lin, Chih-Hao Chen.

**Investigation:** Chao-Wen Lin, Zih-Wei Yang, Chih-Hao Chen, Yu-Wen Cheng.

**Methodology:** Chao-Wen Lin, Chih-Hao Chen.

**Project administration:** Chao-Wen Lin, Chih-Hao Chen.

**Resources:** Chao-Wen Lin, Zih-Wei Yang, Yu-Wen Cheng.

**Supervision:** Chih-Hao Chen, Sung-Chun Tang, Jiann-Shing Jeng.

**Validation:** Chao-Wen Lin, Sung-Chun Tang, Jiann-Shing Jeng.

**Visualization:** Chao-Wen Lin.

**Writing – original draft:** Chao-Wen Lin, Chih-Hao Chen.

**Writing – review & editing:** Chih-Hao Chen, Sung-Chun Tang, Jiann-Shing Jeng.

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
