## [Decision Letter · Decision Letter 0]

26 Apr 2022

PONE-D-22-01077Reduced macular vessel density and inner retinal thickness correlate with the severity of cerebral autosomal dominant arteriopathy with subcortical infarcts and leukoencephalopathy (CADASIL)PLOS ONE

Dear Dr. Chen,

Thank you for submitting your manuscript to PLOS ONE. After careful consideration, we feel that it has merit but does not fully meet PLOS ONE’s publication criteria as it currently stands. Therefore, we invite you to submit a revised version of the manuscript that addresses the points raised during the review process.

We look forward to receiving your revised manuscript.

Kind regards,

Juan Manuel Marquez-Romero, M.D., M.Sc.

Academic Editor

PLOS ONE

Journal Requirements:

Reviewers' comments:

Reviewer's Responses to Questions

**Comments to the Author**

1. Is the manuscript technically sound, and do the data support the conclusions?

Reviewer #1: Yes

2. Has the statistical analysis been performed appropriately and rigorously? 

Reviewer #1: Yes

3. Have the authors made all data underlying the findings in their manuscript fully available?

Reviewer #1: Yes

4. Is the manuscript presented in an intelligible fashion and written in standard English?

Reviewer #1: Yes

5. Review Comments to the Author

Reviewer #1: This manuscript addresses an important need for non-invasive methodologies to diagnose and track the progression of CADASIL. I thought the experiments were well designed and this is the most detailed OCT-A evaluation I have so far seen for this disease population including appropriate controls. The results and shown correlations provide a strong case for the use of this technique for this disease population. In order to improve the paper, I would suggest that some figures to display the correlations (bar graphs) might be helpful when following the text to help readers. I would also suggest a schematic of the eye for the several measured regions to improve clarity. Lastly, the English in the manuscript is good but I would have a native English speaker (if possible) check the text for a few areas where the language isn't clear, particularly in the introduction. For example, CADASIL does not "mainly affect middle-aged adults". It actually affects a wide range of ages but disease onset is generally around the 3rd decade of life.

6. PLOS authors have the option to publish the peer review history of their article (what does this mean?). If published, this will include your full peer review and any attached files.

Reviewer #1: No

---

## [Author Response · Author response to Decision Letter 0]

27 Apr 2022

Response to Reviewer’s Comments

We thank the reviewer for pointing out the suggestions to make the manuscript better. Please kindly see the following point-by-point response. In the revised manuscript, any change or addition of the words, paragraphs, or figures are highlighted in yellow. We hope this revision would substantially strengthen our manuscript.

Reviewer’s Comments: This manuscript addresses an important need for non-invasive methodologies to diagnose and track the progression of CADASIL. I thought the experiments were well designed and this is the most detailed OCT-A evaluation I have so far seen for this disease population including appropriate controls. The results and shown correlations provide a strong case for the use of this technique for this disease population. 

Point 1: In order to improve the paper, I would suggest that some figures to display the correlations (bar graphs) might be helpful when following the text to help readers. 

Response 1: Thank you for the suggestion. We have added a figure (Figure 3) to display the correlations between parameters of OCT-A and cognition, gait, and neuroimaging markers. We used a forest plot to display the multiple correlations estimates and their confidence intervals. Those with significant correlations (p<0.05) were mentioned with text next to it. We hope this figure can be helpful for the readers.

Point 2: I would also suggest a schematic of the eye for the several measured regions to improve clarity. 

Response 2: Thank you for the suggestion. We have added a figure (Figure 1) to show the several pre-defined measured regions of the eye. We hope this figure can also be helpful for the readers.

Point 3: Lastly, the English in the manuscript is good but I would have a native English speaker (if possible) check the text for a few areas where the language isn't clear, particularly in the introduction. For example, CADASIL does not "mainly affect middle-aged adults". It actually affects a wide range of ages but disease onset is generally around the 3rd decade of life. 

Response 3: Thank you for your thoughtful reading. This manuscript has been edited by an Academic Editing company (shown in the following) before submission. Nevertheless, we acknowledged there were still some places for improvement. Therefore, we’ve re-written some parts of the manuscript, including the introduction part. Hope this will improve our manuscript.

---

## [Decision Letter · Decision Letter 1]

3 May 2022

Reduced macular vessel density and inner retinal thickness correlate with the severity of cerebral autosomal dominant arteriopathy with subcortical infarcts and leukoencephalopathy (CADASIL)

PONE-D-22-01077R1

Dear Dr. Chen,

We’re pleased to inform you that your manuscript has been judged scientifically suitable for publication and will be formally accepted for publication once it meets all outstanding technical requirements.

Kind regards,

Juan Manuel Marquez-Romero, M.D., M.Sc.

Academic Editor

PLOS ONE

Reviewers' comments:

Reviewer's Responses to Questions

**Comments to the Author**

1. If the authors have adequately addressed your comments raised in a previous round of review and you feel that this manuscript is now acceptable for publication, you may indicate that here to bypass the “Comments to the Author” section, enter your conflict of interest statement in the “Confidential to Editor” section, and submit your "Accept" recommendation.

Reviewer #1: All comments have been addressed

7. PLOS authors have the option to publish the peer review history of their article (what does this mean?). If published, this will include your full peer review and any attached files.

Reviewer #1: **Yes: **Elisa A Ferrante

---

## [Editor Report · Acceptance letter]

13 May 2022

PONE-D-22-01077R1 

Reduced macular vessel density and inner retinal thickness correlate with the severity of cerebral autosomal dominant arteriopathy with subcortical infarcts and leukoencephalopathy (CADASIL) 

Dear Dr. Chen:

I'm pleased to inform you that your manuscript has been deemed suitable for publication in PLOS ONE. Congratulations! Your manuscript is now with our production department. 

Kind regards, 

on behalf of

Dr. Juan Manuel Marquez-Romero 

Academic Editor

PLOS ONE